# Self-training For Few-shot Transfer Across Extreme Task Differences

**Cheng Perng Phoo, Bharath Hariharan**
Department of Computer Science
Cornell University
{cpphoo, bharathh}@cs.cornell.edu

## Abstract

Most few-shot learning techniques are pre-trained on a large, labeled "base dataset". In problem domains where such large labeled datasets are not available for pre-training (e.g., X-ray, satellite images), one must resort to pre-training in a different "source" problem domain (e.g., ImageNet), which can be very different from the desired target task. Traditional few-shot and transfer learning techniques fail in the presence of such extreme differences between the source and target tasks. In this paper, we present a simple and effective solution to tackle this extreme domain gap: self-training a source domain representation on unlabeled data from the target domain. We show that this improves one-shot performance on the target domain by 2.9 points on average on the challenging BSCD-FSL benchmark consisting of datasets from multiple domains. Our code is available at https://github.com/cpphoo/STARTUP.

## 1 Introduction

Despite progress in visual recognition, training recognition systems for new classes in novel domains requires thousands of labeled training images per class. For example, to train a recognition system for identifying crop types in satellite images, one would have to hire someone to go to the different locations on earth to get the labels of thousands of satellite images. The high cost of collecting annotations precludes many downstream applications.

This issue has motivated research on *few-shot learners*: systems that can *rapidly* learn novel classes from *a few examples*. However, most few-shot learners are trained on a large *base dataset* of classes from the same domain. This is a problem in many domains (such as medical imagery, satellite images), where no large labeled dataset of base classes exists. The only alternative is to train the few-shot learner on a different domain (a common choice is to use ImageNet). Unfortunately, few-shot learning techniques often assume that novel and base classes share modes of variation (Wang et al., 2018), class-distinctive features (Snell et al., 2017), or other inductive biases. These assumptions are broken when the difference between base and novel is as extreme as the difference between object classification in internet photos and pneumonia detection in X-ray images. As such, recent work has found that all few-shot learners fail in the face of such extreme task/domain differences, underperforming even naive transfer learning from ImageNet (Guo et al., 2020).

Another alternative comes to light when one considers that many of these problem domains have *unlabeled data* (e.g., undiagnosed X-ray images, or unlabeled satellite images). This suggests the possibility of using *self-supervised techniques* on this unlabeled data to produce a good feature representation, which can then be used to train linear classifiers for the target classification task using just a few labeled examples. Indeed, recent work has explored self-supervised learning on a variety of domains (Wallace & Hariharan, 2020). However, self-supervised learning starts *tabula rasa*, and as such requires extremely large amounts of unlabeled data (on the order of millions of images). With more practical unlabeled datasets, self-supervised techniques still struggle to outcompete naive ImageNet transfer (Wallace & Hariharan, 2020). We are thus faced with a conundrum: on the one hand, few-shot learning techniques fail to bridge the extreme differences between ImageNet and domains such as X-rays. On the other hand, self-supervised techniques fail when they ignore inductive biases from ImageNet. A sweet spot in the middle, if it exists, is elusive.

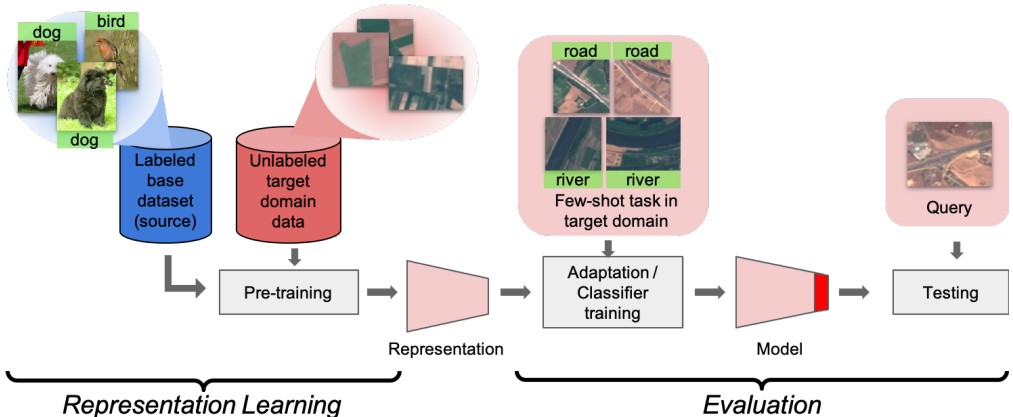

Figure 1: Problem setup. In the representation learning phase (left), the learner has access to a large labeled "base dataset" in the source domain, and some unlabeled data in the target domain, on which to pre-train its representation. The learner must then rapidly learn/adapt to few-shot tasks in the target domain in the evaluation phase (right).

In this paper, we solve this conundrum by presenting a strategy that adapts feature representations trained on source tasks to *extremely different* target domains, so that target task classifiers can then be trained on the adapted representation with very little labeled data. Our key insight is that a pre-trained base classifier from the source domain, when applied to the target domain, induces a *grouping* of images on the target domain. This grouping captures what the pre-trained classifier thinks are similar or dissimilar in the target domain. Even though the classes of the pre-trained classifier are themselves irrelevant in the target domain, the induced notions of *similarity and dissimilarity* might still be relevant and informative. This induced notion of similarity is in contrast to current self-supervised techniques which often function by considering each image as its own class and dissimilar from every other image in the dataset (Wu et al., 2018; Chen et al., 2020). We propose to train feature representations on the novel target domain to *replicate this induced grouping*. This approach produces a feature representation that is (a) adapted to the target domain, while (b) maintaining prior knowledge from the source task to the extent that it is relevant. A discerning reader might observe the similarity of this approach to *self-training*, except that our goal is to adapt the feature representation to the target domain, rather than improve the base classifier itself.

We call our approach "**S**elf **T**raining to **A**dapt **R**epresentations **T**o **U**nseen **P**roblems", or STARTUP. In a recently released BSCD-FSL benchmark consisting of datasets from extremely different domains (Guo et al., 2020), we show that STARTUP provides significant gains (up to 2.9 points on average) over few-shot learning, transfer learning and self-supervision state-of-the-art. To the best of our knowledge, ours is the first attempt to bridge such large task/domain gaps and successfully and consistently outperform naive transfer in cross-domain few-shot learning.

## 2 PROBLEM SETUP

Our goal is to build learners for novel domains that can be *quickly* trained to recognize new classes when presented with *very few* labeled data points (*"few-shot"*). Formally, the target domain is defined by a set of data points (e.g. images) $\mathcal{X}_\mathcal{N}$, an unknown set of classes (or label space) $\mathcal{Y}_\mathcal{N}$, and a distribution $\mathcal{D}_\mathcal{N}$ over $\mathcal{X}_\mathcal{N} \times \mathcal{Y}_\mathcal{N}$. A "few-shot learning task" in this domain will consist of a set of classes $Y \subset \mathcal{Y}_\mathcal{N}$, a very small training set ("support")

$$S = \{(x_i, y_i)\}_{i=1}^n \sim \mathcal{D}_\mathcal{N}^n, \quad y_i \in Y$$

and a small test set ("query")

$$Q = \{x_i\}_{i=1}^m \sim \mathcal{D}_\mathcal{N}^m$$

When presented with such a few-shot learning task, the learner must rapidly learn the classes presented and accurately classify the query images.

As with prior few-shot learning work, we will assume that before being presented with few-shot learning tasks in the target domain, the learner has access to a large annotated dataset $D_B$ known as the base dataset. However, crucially *unlike prior work on few-shot learning*, we assume that this base dataset is drawn from a very different distribution. In fact, we assume that the base dataset is drawn from a completely disjoint image space $\mathcal{X}_B$ and a disjoint set of classes $\mathcal{Y}_B$:

$$D_B = \{(x_i, y_i)\}_{i=1}^{N_B} \subset \mathcal{X}_{\mathcal{B}} \times \mathcal{Y}_{\mathcal{B}}$$

where $\mathcal{X}_{\mathcal{B}}$ is the set of data (or the source domain) and $\mathcal{Y}_B$ is the set of base classes. Because the base dataset is so different from the target domain, we introduce another difference vis-a-vis the conventional few-shot learning setup: the learner is given access to an additional unlabeled dataset from the target domain:

$$D_u = \{x_i\}_{i=1}^{N_u} \sim \mathcal{D}_{\mathcal{N}}^{N_u}$$

Put together, the learner will undergo two phases. In the *representation learning* phase, the learner will pre-train its representation on $D_B$ and $D_u$; then it goes into the *evaluation* phase where it will be presented few-shot tasks from the target domain where it learns the novel classes (Figure 1).

## 3    RELATED WORK

**Few-shot Learning (FSL).** This paper explores few-shot transfer, and as such the closest related work is on few-shot learning. Few-shot learning techniques are typically predicated on some degree of similarity between classes in the base dataset and novel classes. For example, they may assume that features that are discriminative for the base classes are also discriminative for the novel classes, suggesting a metric learning-based approach (Gidaris & Komodakis, 2018; Qi et al., 2018; Snell et al., 2017; Vinyals et al., 2016; Sung et al., 2018; Hou et al., 2019) or transfer learning-based approach (Chen et al., 2019b; Wang et al., 2019; Kolesnikov et al., 2020; Tian et al., 2020). Alternatively, they may assume that model initializations that lead to rapid convergence on the base classes are also good initializations for the novel classes (Finn et al., 2017; 2018; Ravi & Larochelle, 2017; Nichol & Schulman; Rusu et al., 2019; Sun et al., 2019; Lee et al., 2019). Other methods assume that modes of intra-class variation are shared, suggesting the possibility of learned, class-agnostic augmentation policies (Hariharan & Girshick, 2017; Wang et al., 2018; Chen et al., 2019c). Somewhat related is the use of a class-agnostic parametric model that can "denoise" few-shot models, be they from the base or novel classes (Gidaris & Komodakis, 2018; 2019). In contrast to such strong assumptions of similarity between base and novel classes, this paper tackles few-shot learning problems where base and novel classes come from very different domains, also called cross-domain few-shot learning.

**Cross-domain Few-shot Classification (CD-FSL).** When the domain gap between the base and novel dataset is large, recent work (Guo et al., 2020; Chen et al., 2019b) has shown that existing state-of-the-art few-shot learners fail to generalize. Tseng et al. (2020) attempt to address this problem by simulating cross-domain transfer during training. However, their approach assumes access to an equally diverse array of domains during training, and a much smaller domain gap at test time: for example, both base and novel datasets are from internet images. Another relevant work (Ngiam et al., 2018) seeks to build domain-specific feature extractor by reweighting different classes of examples in the base dataset based on the target novel dataset but their work only investigates transfer between similar domains (both source and target are internet images). Our paper tackles a more extreme domain gap. Another relevant benchmark for this problem is (Zhai et al., 2019) but they assume access to more annotated examples (1k annotations) during test time than the usual FSL setup.

**Few-shot learning with unlabeled data.** This paper uses unlabeled data from the target domain to bridge the domain gap. Semi-supervised few-shot learning (SS-FSL) (Ren et al., 2018; Li et al., 2019; Yu et al., 2020; Rodríguez et al., 2020; Wang et al., 2020) and transductive few-shot learning (T-FSL) (Liu et al., 2019; Dhillon et al., 2020; Hou et al., 2019; Wang et al., 2020; Rodríguez et al., 2020) do use such unlabeled data, but only during evaluation, assuming that representations trained on the base dataset are good enough. In contrast our approach leverages the unlabeled data during representation learning. The two are orthogonal innovations and can be combined.

**Self-Training.** Our approach is closely related to self-training, which has been shown to be effective for semi-supervised training and knowledge distillation. In self-training, a teacher model trained on

the labeled data is used to label the unlabeled data and another student model is trained on both the original labeled data and the unlabeled data labeled by the teacher. Xie et al. (2020) and Yalniz et al. (2019) have shown that using self-training can improve ImageNet classification performance. Knowledge distillation (Hinton et al., 2015) is similar but aims to compress a large teacher network by training a student network to mimic the prediction of the teacher network. A key difference between these and our work is that self-training / knowledge distillation focus on a single task of interest, i.e, there is no change in label space. Our approach is similar, but we are interested in transferring to novel domains with a *wholly different label space*: an unexplored scenario.

**Domain Adaptation.** Transfer to new domains is also in the purview of domain adaptation (Tzeng et al., 2017; Hoffman et al., 2018; Long et al., 2018; Xu et al., 2019; Laradji & Babanezhad, 2020; Wang & Deng, 2018; Wilson & Cook, 2020) where the goal is to transfer knowledge from the label-abundant source domain to a target domain where only unlabeled data is available. In this realm, self-training has been extensively explored (Zou et al., 2018; Chen et al., 2019a; Zou et al., 2019; Zhang et al., 2019; Mei et al., 2020). However, a key assumption in domain adaptation is that the source domain and target domain share the same label space which does not hold for FSL.

**Self-supervised Learning.** Learning from unlabeled data has seen a resurgence of interest with advances in self-supervised learning. Early self-supervised approaches were based on handcrafted "pretext tasks" such as solving jigsaw puzzles (Noroozi & Favaro, 2016), colorization (Zhang et al., 2016) or predicting rotation (Gidaris et al., 2018). A more recent (and better performing) line of self-supervised learning is contrastive learning (Wu et al., 2018; Misra & Maaten, 2020; He et al., 2020; Chen et al., 2020) which aims to learn representations by considering each image together with its augmentations as a separate class. While self supervision has been shown to boost few-shot learning (Gidaris et al., 2019; Su et al., 2020), its utility in cases of large domain gaps between base and novel datasets have not been evaluated. Our work focuses on this challenging scenario.

## 4 APPROACH

Consider a classification model $f_\theta = C \circ \phi$ where $\phi$ embeds input $x$ into $\mathbb{R}^d$ and $C$ is a (typically linear) classifier head that maps $\phi(x)$ to predicted probabilities $P(y|x)$. $\theta$ is a vector of parameters. During representation learning, STARTUP performs the following three steps:

1. Learn a teacher model $\theta_0$ on the base dataset $\mathcal{D}_B$ by minimizing the cross entropy loss

2. Use the teacher model to construct a softly-labeled set $\mathcal{D}_u^* = \{(x_i, \bar{y}_i)\}_{i=1}^{N_u}$ where

$$\bar{y}_i = f_{\theta_0}(x_i) \quad \forall x_i \in \mathcal{D}_u. \tag{1}$$

   Note that $\bar{y}_i$ is a probability distribution as described above.

3. Learn a new student model $\theta^*$ on $\mathcal{D}_B$ and $\mathcal{D}_u^*$ by optimizing:

$$\min_\theta \frac{1}{N_B} \sum_{(x_i, y_i) \in \mathcal{D}_B} l_{CE}(f_\theta(x_i), y_i) + \frac{1}{N_u} \sum_{(x_j, \bar{y}_j) \in \mathcal{D}_u^*} l_{KL}(f_\theta(x_j), \bar{y}_j) + l_{unlabeled}(\mathcal{D}_u) \tag{2}$$

   where $l_{CE}$ is the cross entropy loss, $l_{KL}$ is the KL divergence and $l_{unlabeled}$ is any unsupervised/self-supervised loss function (See below).

The third term, $l_{unlabeled}$, is intended to help the learner extract additional useful knowledge specific to the target domain. We use a state-of-the-art self-supervised loss function based on contrastive learning: *SimCLR* (Chen et al., 2020). The SimCLR loss encourages two augmentations of the same image to be closer in feature space to each other than to other images in the batch. We refer the reader to the paper for the detailed loss formulation.

The first two terms are similar to those in prior self-training literature (Xie et al., 2020). However, while in prior self-training work, the second term ($l_{KL}$) is thought to mainly introduce noise during training, we posit that $l_{KL}$ has a more substantial role to play here: it encourages the model to learn feature representations that *emphasize* the groupings induced by the pseudo-labels $\bar{y}_i$ on the target domain. We analyze this intuition in section 5.2.2.

### 4.1 EVALUATION

STARTUP is agnostic to inference methods during evaluation; any inference methods that rely on a representation (Snell et al., 2017; Gidaris & Komodakis, 2018) can be used with STARTUP. For simplicity and based on results reported by Guo et al. (2020), we freeze the representation $\phi$ after performing STARTUP and train a linear classifier on the support set and evaluate the classifier on the query set.

### 4.2 INITIALIZATION STRATEGIES

Xie et al. (2020) found that training the student from scratch sometimes yields better results for ImageNet classification. To investigate, we focused on a variant of STARTUP where the SimCLR loss is omitted and experimented with three different initialization strategies - from scratch (STARTUP-Rand (no SS)), from teacher (STARTUP-T (no SS)) and using the teacher's embedding with randomly initialized classifier (STARTUP (no SS)). We found no conclusive evidence that one single initialization strategy is superior to the others across different datasets (See Appendix A.4) but we observe that (STARTUP (no SS)) is either the best or the second best in all scenarios. As such, we opt to use teacher's embedding with a randomly initialized classifier as the default student initialization.

## 5 EXPERIMENTS

We defer the implementation details to Appendix A.1.

### 5.1 FEW-SHOT TRANSFER ACROSS DRASTICALLY DIFFERENT DOMAINS

**Benchmark.** We experiment with the challenging (BSCD-FSL) benchmark introduced in Guo et al. (2020). The *base dataset* in this benchmark is miniImageNet (Vinyals et al., 2016), which is an object recognition task on internet images. There are 4 novel datasets in the benchmark, none of which involve objects, and all of which come from a very different domain than internet images: CropDiseases (recognizing plant diseases in leaf images), EuroSAT (predicting land-use from satellite images), ISIC2018 (identifying melanoma from images of skin lesions) and ChestX (diagnosing chest X-rays). Guo et al. found that state-of-the-art few-shot learners fail on this benchmark.

To construct our setup, we randomly sample 20% of data from each novel datasets to form the respective unlabeled datasets $\mathcal{D}_u$. We use the rest for sampling tasks for evaluation. Following Guo et al. (2020), we evaluate 5-way k-shot classification tasks (the support set consists of 5 classes and k examples per class) for $k \in \{1, 5\}$ and report the mean and 95% confidence interval over 600 few-shot tasks (conclusions generalize to $k \in \{20, 50\}$. See Appendix A.2).

**Baselines.** We compare to the techniques reported in Guo et al. (2020), which includes most state-of-the-art approaches as well as a cross-domain few-shot technique Tseng et al. (2020). The top performing among these is naive **Transfer** which simply trains a convolutional network to classify the base dataset, and uses the resulting representation to learn a linear classifier when faced with novel few-shot tasks. These techniques do not use the novel domain unlabeled data.

We also compare to another baseline, **SimCLR** that uses the novel domain unlabeled data $D_u$ to train a representation using SimCLR(Chen et al., 2020), and then uses the resulting representation to learn linear classifiers for few-shot tasks. This builds upon state-of-the-art self-supervised techniques.

To compare to a baseline that uses both sources of data, we establish **Transfer + SimCLR**. This baseline is similar to the SimCLR baseline except the embedding is initialized to Transfer's embedding before SimCLR training.

Following the benchmark, all methods use a ResNet-10 (He et al., 2016) unless otherwise stated.

### 5.1.1 RESULTS

We present our main results on miniImageNet $\rightarrow$ BSCD-FSL in Table 1.

**STARTUP vs Few-shot learning techniques.** STARTUP performs significantly better than all few-shot techniques in most datasets (except ChestX, where all methods are similar). Compared to

Table 1: 5-way k-shot classification accuracy on miniImageNet→BSCD-FSL. Mean and 95% confidence interval are reported. (no SS) indicates removal of SimCLR. ProtoNet: (Snell et al., 2017), MAML:(Finn et al., 2017), MetaOpt:(Lee et al., 2019) FWT:(Tseng et al., 2020). *Numbers reported in (Guo et al., 2020); our re-implementation of Transfer uses a different batch size and 80% of the original test set for evaluation. Bolded entries are top performing methods that are not different based on t-test at significant level 0.05.

| Methods | ChestX | | ISIC | |
|---|---|---|---|---|
| | k=1 | k=5 | k=1 | k=5 |
| MAML* | - | $23.48 \pm 0.96$ | - | $40.13 \pm 0.58$ |
| ProtoNet* | - | $24.05 \pm 1.01$ | - | $39.57 \pm 0.57$ |
| ProtoNet + FWT* | - | $23.77 \pm 0.42$ | - | $38.87 \pm 0.52$ |
| MetaOpt* | - | $22.53 \pm 0.91$ | - | $36.28 \pm 0.50$ |
| Transfer* | - | $25.35 \pm 0.96$ | - | $43.56 \pm 0.60$ |
| Transfer | $22.71 \pm 0.40$ | $\mathbf{26.71 \pm 0.46}$ | $30.71 \pm 0.59$ | $43.08 \pm 0.57$ |
| SimCLR | $22.10 \pm 0.41$ | $25.02 \pm 0.42$ | $26.25 \pm 0.53$ | $36.09 \pm 0.57$ |
| Transfer + SimCLR | $22.70 \pm 0.40$ | $\mathbf{26.95 \pm 0.45}$ | $\mathbf{32.63 \pm 0.63}$ | $45.96 \pm 0.61$ |
| STARTUP (no SS) | $\mathbf{22.87 \pm 0.41}$ | $\mathbf{26.68 \pm 0.45}$ | $\mathbf{32.24 \pm 0.62}$ | $46.48 \pm 0.61$ |
| STARTUP | $\mathbf{23.09 \pm 0.43}$ | $\mathbf{26.94 \pm 0.44}$ | $\mathbf{32.66 \pm 0.60}$ | $\mathbf{47.22 \pm 0.61}$ |
| Methods | EuroSAT | | CropDisease | |
| | k=1 | k=5 | k=1 | k=5 |
| MAML* | - | $71.70 \pm 0.72$ | - | $78.05 \pm 0.68$ |
| ProtoNet* | - | $73.29 \pm 0.71$ | - | $79.72 \pm 0.67$ |
| ProtoNet + FWT* | - | $67.34 \pm 0.76$ | - | $72.72 \pm 0.70$ |
| MetaOpt* | - | $64.44 \pm 0.73$ | - | $68.41 \pm 0.73$ |
| Transfer* | - | $75.69 \pm 0.66$ | - | $87.48 \pm 0.58$ |
| Transfer | $60.73 \pm 0.86$ | $80.30 \pm 0.64$ | $69.97 \pm 0.85$ | $90.16 \pm 0.49$ |
| SimCLR | $43.52 \pm 0.88$ | $59.05 \pm 0.70$ | $\mathbf{78.23 \pm 0.83}$ | $92.57 \pm 0.48$ |
| Transfer + SimCLR | $57.18 \pm 0.87$ | $77.61 \pm 0.66$ | $76.90 \pm 0.78$ | $92.64 \pm 0.44$ |
| STARTUP (no SS) | $62.90 \pm 0.83$ | $81.81 \pm 0.61$ | $73.30 \pm 0.82$ | $91.69 \pm 0.47$ |
| STARTUP | $\mathbf{63.88 \pm 0.84}$ | $\mathbf{82.29 \pm 0.60}$ | $75.93 \pm 0.80$ | $\mathbf{93.02 \pm 0.45}$ |

previous state-of-the-art, **Transfer**, we observe an average of 2.9 points improvement on the 1-shot case. The improvement is particularly large on CropDisease, where STARTUP provides almost a 6 point increase for 1-shot classification. This improvement is significant given the simplicity of our approach, and given that all meta-learning techniques *underperform* this baseline.

**STARTUP vs SimCLR.** The SimCLR baseline in general tends to *underperform* naive transfer from miniImageNet, and consequently, STARTUP performs significantly better than SimCLR on ISIC and EuroSAT. The exception to this is CropDisease, where SimCLR produces a surprisingly good representation. We conjecture that the base embedding is not a good starting point for this dataset. However, we find that using SimCLR as an auxilliary loss to train the student (STARTUP vs STARTUP (no SS)) is beneficial.

**STARTUP vs Transfer + SimCLR.** STARTUP outperforms Transfer + SimCLR in most cases (except 5-shot in ChestX and 1-shot in ISIC). We stress that the strength of STARTUP is not solely from SimCLR but rather from both self-training and SIMCLR. This is especially evident in EuroSAT since the STARTUP (no SS) variant outperforms **Transfer** and **Transfer + SimCLR**.

**Larger and stronger teachers.** To unpack the impact of teacher quality, we experiment with a larger network and transfer from the full ILSVRC 2012 dataset (Deng et al., 2009) to BSCD-FSL.

Table 2: 5-way k-shot classification accuracy on ImageNet(ILSVRC 2012)→BSCD-FSL. Bolded entries are top performing methods that are not different based on t-test at significant level 0.05.

| Methods | ChestX | | ISIC | |
|---|---|---|---|---|
| | k=1 | k=5 | k=1 | k=5 |
| Transfer | $21.97 \pm 0.39$ | $25.85 \pm 0.41$ | $30.27 \pm 0.51$ | $43.88 \pm 0.56$ |
| STARTUP (no SS) | $\mathbf{22.90 \pm 0.40}$ | $26.74 \pm 0.46$ | $30.18 \pm 0.56$ | $44.19 \pm 0.57$ |
| STARTUP | $\mathbf{23.03 \pm 0.42}$ | $\mathbf{27.24 \pm 0.46}$ | $\mathbf{31.69 \pm 0.59}$ | $\mathbf{46.02 \pm 0.59}$ |
| Methods | EuroSAT | | CropDisease | |
| | k=1 | k=5 | k=1 | k=5 |
| Transfer | $66.08 \pm 0.81$ | $85.58 \pm 0.48$ | $74.17 \pm 0.82$ | $92.46 \pm 0.42$ |
| STARTUP (no SS) | $70.08 \pm 0.80$ | $87.12 \pm 0.45$ | $80.13 \pm 0.77$ | $94.51 \pm 0.38$ |
| STARTUP | $\mathbf{73.83 \pm 0.77}$ | $\mathbf{89.70 \pm 0.41}$ | $\mathbf{85.10 \pm 0.74}$ | $\mathbf{96.06 \pm 0.33}$ |

In particular, we used the publicly available pre-trained ResNet-18 (He et al., 2016) as a teacher and train a student via STARTUP. We compare this to a transfer baseline that uses the same network and ImageNet as the training set. The result can be found in table 2. Surprisingly, larger, richer embeddings *do not always transfer better*, in contrast to in-domain results reported by Hariharan & Girshick (2017). However, STARTUP is still useful in improving performance: the absolute improvement in performance for STARTUP compared to Transfer remains about the same in most datasets except EuroSAT and CropDisease where larger improvements are observed.

## 5.2 WHY SHOULD STARTUP WORK?

While it is clear that STARTUP helps improve few shot transfer across extreme domain differences, it is not clear why or how it achieves this improvement. Below, we look at a few possible hypotheses.

### 5.2.1 HYPOTHESIS 1: STARTUP ADDS NOISE WHICH INCREASES ROBUSTNESS.

Xie et al. (2020) posit that self-training introduces noise when training the student and thus yielding a more robust student. More robust students may be learning more generalizable representations, and this may be allowing STARTUP to bridge the domain gap. Under this hypothesis, the function of the unlabeled data is only to add noise during training. This in turn suggests that STARTUP should yield improvements on the target tasks *even if trained on unlabeled data from a different domain*. To test this, we train a STARTUP ResNet-18 student on EuroSAT and ImageNet and evaluate it on CropDisease. This model yields a 5-way 1-shot performance of $70.40 \pm 0.86$ ($88.78 \pm 0.54$ for 5-shot), significantly *underperforming* the naive Transfer baseline (Table 2. See Appendix A.7 for different combinations of unlabeled dataset and target dataset). This suggests that **while the hypothesis is valid in conventional semi-supervised learning, it is incorrect in the cross-domain few-shot learning setup: unlabeled data are not merely functioning as noise**. Rather, STARTUP is learning inherent structure in the target domain useful for downstream classification. The question now becomes what inherent structure STARTUP is learning, which leads us to the next hypothesis.

### 5.2.2 HYPOTHESIS 2: STARTUP ENHANCES TEACHER-INDUCED GROUPINGS

**The teacher produces a meaningful grouping of the data from the target domain.** The predictions made by the teacher essentially induce a grouping on the target domain. Even though the base label space and novel label space are disjoint, the groupings produced by the teacher might not be entirely irrelevant for the downstream classification task. To test this, we first assign each example in the novel datasets to its most probable prediction by the teacher (ResNet 18 trained on ImageNet). We then compute the adjusted mutual information (AMI) (Vinh et al., 2010) between the resulting grouping and ground truth label. AMI ranges from 0 for unrelated groupings to 1 for identical groupings. From Table 3, we see that on EuroSAT and CropDisease, there is quite a bit of agree-

Table 3: Adjusted Mutual Information (AMI) of the grouping induced by the teacher and the ground truth label. AMI has value from 0 to 1 with higher value indicating more agreement.

|  | ChestX | ISIC | EuroSAT | CropDisease |
|---|---|---|---|---|
| AMI | 0.0075 | 0.0427 | 0.3079 | 0.2969 |

Figure 2: t-SNE plot of EuroSAT and CropDisease prior to and after STARTUP.

ment between the induced grouping and ground truth label. Interestingly, these are the two datasets where we observe the best transfer performance and most improvement from STARTUP (Table 2), suggesting correlations between the agreement and the downstream classification task performance.

**STARTUP enhances the grouping induced by the teacher.** Even though the induced groupings by the teacher can be meaningful, one could argue that those groupings are captured in the teacher model already, and no further action to update the representation is necessary. However, we posit that STARTUP encourages the feature representations to emphasize the grouping. To verify, we plot the t-SNE (Maaten & Hinton, 2008) of the data prior to STARTUP and after STARTUP for the two datasets in figure 2. From the t-SNE plot, we observe more separation after doing STARTUP, signifying a representation with stronger discriminability.

Put together, this suggests that **STARTUP works by (a) inducing a potentially meaningful grouping on the target domain data, and (b) training a representation that emphasizes this grouping.**

### 5.3 FEW-SHOT TRANSFER ACROSS SIMILAR DOMAINS

Is STARTUP still useful when the gap between the base and target is smaller? To answer this, we tested STARTUP on two popular within-domain few-shot learning benchmark: miniImageNet (Vinyals et al., 2016) and tieredImageNet (Ren et al., 2018). For miniImageNet, we use 20% of the novel set as the unlabeled dataset and use the same teacher as in section 5.1. For tieredImageNet, we use ResNet-12 (Oreshkin et al., 2018) as our model architecture and evaluate two different setups - tieredImageNet-less that uses 10% of the novel set as unlabeled data (following Ren et al. (2018)) and tieredImageNet-more that uses 50% of the novel set as unlabeled data. We follow the same evaluation protocols in section 5.1.

We report the results in table 4. We found that on miniImageNet, STARTUP and its variants neither helps nor hurts in most cases (compared to **Transfer**), indicating that the representation is already well-matched. On both variants of tieredImageNet, we found that STARTUP, with the right initialization, can in fact outperform **Transfer**. In particular, in the less data case, it is beneficial to initialize the student with the teacher model whereas in the more data case, training the students from scratch is superior. In sum, these results show the potential of STARTUP variants to boost few-shot transfer even when the base and target domains are close.

Table 4: 5-way 1-shot (top) and 5-way 5-shot (bottom) classification accuracy on miniImagenet and tieredImageNet. Bolded entries are top performing methods that are not different based on t-test at significant level 0.05.

| Methods (k=1) | miniImageNet | tieredImageNet-less | tieredImageNet-more |
|---|---|---|---|
| Transfer | **54.18 ± 0.79** | 57.29 ± 0.83 | 57.68 ± 0.89 |
| STARTUP-T (no SS) | **53.91 ± 0.79** | **60.39 ± 0.86** | 61.00 ± 0.86 |
| STARTUP (no SS) | 53.74 ± 0.80 | 55.49 ± 0.85 | 57.19 ± 0.89 |
| STARTUP-Rand (no SS) | **54.15 ± 0.81** | 56.93 ± 0.91 | **63.29 ± 0.90** |
| STARTUP-T | **54.00 ± 0.80** | **60.19 ± 0.86** | 60.16 ± 0.86 |
| STARTUP | **54.20 ± 0.81** | 55.33 ± 0.85 | 53.88 ± 0.89 |
| STARTUP-Rand | **53.89 ± 0.87** | 56.93 ± 0.91 | 61.95 ± 0.93 |
| Methods (k=5) | miniImageNet | tieredImageNet-less | tieredImageNet-more |
| Transfer | 76.20 ± 0.64 | 79.05 ± 0.65 | 78.67 ± 0.69 |
| STARTUP-T (no SS) | **76.26 ± 0.64** | **80.14 ± 0.65** | 79.61 ± 0.68 |
| STARTUP (no SS) | **76.42 ± 0.63** | 78.36 ± 0.66 | 78.50 ± 0.68 |
| STARTUP-Rand (no SS) | 73.77 ± 0.66 | **79.58 ± 0.69** | **81.60 ± 0.66** |
| STARTUP-T | 76.21 ± 0.63 | 79.40 ± 0.67 | 79.04 ± 0.68 |
| STARTUP | **76.48 ± 0.63** | 77.78 ± 0.67 | 77.48 ± 0.68 |
| STARTUP-Rand | 71.08 ± 0.72 | **79.58 ± 0.69** | 81.03 ± 0.66 |

**Additional Ablation Studies:** We conducted three additional ablation studies: (a) training the student with various amount of unlabeled data, (b) training the student without the base dataset and (c) using the rotation as self-supervision instead of SimCLR in STARTUP . We show that STARTUP benefits from more unlabeled data (Appendix A.5), training student without the base dataset can hurt performance in certain datasets but not all datasets (Appendix A.6) and STARTUP (w/ Rotation) outperforms **Transfer** in certain datasets but underperforms its SimCLR counterparts (Appendix A.3).

## 6 CONCLUSION

We investigate the use of unlabeled data from novel target domains to mitigate the performance degradation of few-shot learners due to large domain/task differences. We introduce STARTUP - a simple yet effective approach that allows few-shot learners to adapt feature representations to the target domain while retaining class grouping induced by the base classifier. We show that STARTUP outperforms prior art on extreme cross-domain few-shot transfer.

## 7 ACKNOWLEDGEMENT

This work is funded by the DARPA LwLL program.

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

# A  APPENDIX

## A.1  IMPLEMENTATION DETAILS

We implemented STARTUP by modifying the the publicly-available implementation [1] of BSCD-FSL by Guo et al. (2020).

### A.1.1  TRAINING THE TEACHER

1. MiniImageNet: We train the teacher model using the code provided in the BSCD-FSL benchmark. We keep everything the same except setting the batch size from 16 to 256.

2. TieredImageNet: We used the same setup as miniImageNet except we reduce the number of epochs to 90. We do not use any image augmentation for tieredImageNet.

3. ImageNet: We used the pretrained ResNet18 available on PyTorch (Paszke et al., 2019)

### A.1.2  TRAINING THE STUDENT

**Optimization Details.** Regardless of the base and novel datasets, the student model is trained for 1000 epochs where an epoch is defined to be a complete pass on the unlabeled data. We use a batch size of 256 on the unlabeled dataset and a batch size of 256 for the base dataset if applicable. We use the SGD with momentum optimizer with momentum 0.9 and weight decay 1e-4. To pick the suitable starting learning rate, 10% of the unlabeled data and 5% of the labeled data (1% when using ImageNet as the base dataset) are set aside as our internal validation set. We pick the starting learning rate by training the student with starting learning rate $lr \in \{$1e-1, 5e-2, 3e-2, 1e-2, 5e-3, 3e-3, 1e-3$\}$ for k epochs where k is the smallest epoch that guarantees at least 50 updates to the model and pick the learning rate that yields lowest loss on the validation set as the starting learning rate. We reduce the learning rate by a factor of 2 when the training loss has not decreased by 20 epochs. The model that achieves the lowest loss on the internal validation set throughout the 1000 epochs of training is picked as the final model.

**SimCLR.** Our implementation of SimCLR's loss function is based on a publicly available implementation of SimCLR [2]. We added the two-layer projection head on top of the embedding function $\phi$. The temperature of NT-Xent is set to 1 since there is no validation set for BSCD-FSL for hyperparameter selection and we use a temperature of 1 when inferring the soft label of the unlabeled set. For the stochastic image augmentations for SimCLR, we use the augmentations defined for each novel dataset in Guo et al. (2020). These augmentations include the commonly used "randomly resized crop", color jittering, random horizontal flipping. For tieredImageNet and miniImageNet, we use the stochastic transformation implemented for the BSCD-FSL benchmark. We refer readers to the BSCD-FSL implementation for more details.

When training the student on the base dataset, we use the augmentation used for training the teacher for fair comparison. The batchsize for SIMCLR is set to 256.

### A.1.3  TRAINING LINEAR CLASSIFIER.

We use the implementation by BSCD-FSL, i.e training the linear classifier with standard cross entropy and SGD optimizer. The linear classifier is trained for 100 epochs with learning rate 0.01, momentum 0.9 and weight decay 1e-4.

### A.1.4  BASELINES

We use the same evaluation methods - linear classifier. Please see A.1.3 for classifier training.

**Transfer.** This is implemented using the teacher model as feature extractor. Please see A.1.1 for details.

**SimCLR.** This is implemented similarly to the SimCLR loss described in A.1.2

---

[1]https://github.com/IBM/cdfsl-benchmark
[2]https://github.com/sthalles/SimCLR

### A.1.5 T-SNE

We use the publicly available scikit-learn implementation of t-SNE (Buitinck et al., 2013). We used the default parameters except for the perplexity where we set to 50. To speed up the experiment, we randomly sampled 25% of the data used for sampling few-shot tasks (80 % of the full dataset) and run t-SNE on this subset.

### A.2 FULL RESULTS ON BSCD-FSL

We present the result on miniImageNet $\to$ BSCD-FSL for shot = 1, 5, 20, 50 in Table 5 and 6. In addition to STARTUP, we also reported results on using teacher model as student initialization (STARTUP-T and STARTUP-T (no SS)) in these tables for reference. Results on ImageNet $\to$ BSCD-FSL can be found in Table 7 and 8. The conclusions we found in 5.1 still hold for higher shots in general.

### A.3 USING ROTATION FOR SELF-SUPERVISION.

We use rotation (Gidaris et al. (2018)) instead of SimCLR in STARTUPand report the results in in Table 5 and 6. We observe that STARTUP (w/ Rotation) is able to outperform **Transfer** in CropDisease and EuroSAT but generally underperforms its SimCLR counterparts.

### A.4 INITIALIZATION STRATEGIES FOR THE STUDENT

We investigate the impact of different initialization strategies for the student on STARTUP. For this experiment, we remove SimCLR from STARTUP and consider three initialization strategies for the student - from scratch (STARTUP-Rand (no SS)), from teacher embedding with a randomly initialized classifier (STARTUP (no SS)), from teacher model (STARTUP-T (no SS)). We repeated the experiment in section 5.1 on miniImageNet $\to$ BSCD-FSL and report the results in table 9. We found that not a single initialization is superior to the others (for instance random initialization is the best on CropDisease but the worst on ISIC) however we did find that initializing the student with the teacher's embedding with a randomly initialized classifier for STARTUP is either the best or second best in all scenarios so we set that as our default initialization.

### A.5 IMPACT OF DIFFERENT AMOUNT OF UNLABELED EXAMPLES

STARTUP uses unlabeled data to adapt feature representations to novel domains. As with all learning techniques, it should perform better with more unlabeled data. To investigate how the amount of unlabeled examples impacts STARTUP, we repeated the miniImageNet $\to$ ISIC experiments in 5.1 with various amount of unlabeled data (20% of the dataset (2003 examples) is set aside for evaluation). The verdict is clear - STARTUP benefits from more unlabeled data (Figure 3).

### A.6 TRAINING THE STUDENT WITHOUT THE BASE DATASET

STARTUP requires joint training on both the base dataset as well as the target domain. But in many cases, the base dataset may not be available. Removing the cross entropy loss on the base dataset when training the student essentially boils down to a fine-tuning paradigm. For miniImageNet $\to$ BSCD-FSL (Table 10), we found no discernible difference between all datasets except on the ISIC where we observe significant degradation in 5-shot performance.

### A.7 STARTUP ON DIFFERENT UNLABELED DATA

We consider the ImageNet $\to$ CD-FSL experiment. We perform STARTUP on unlabeled data different from the target domain and present the result in Table 11. We found that it is crucial that the unlabeled data to perform STARTUP on should be from the target domain of interest.

Table 5: 5-way k-shot classification accuracy on miniImageNet→BSCD-FSL. Mean and 95% confidence interval are reported. (no SS) indicates removal of SimCLR. ProtoNet: (Snell et al., 2017), MAML:(Finn et al., 2017), MetaOpt:(Lee et al., 2019), FWT:(Tseng et al., 2020). *Numbers reported in (Guo et al., 2020); our re-implementation of Transfer uses a different batch size and 80% of the original test set for evaluation. Bolded entries are top performing methods that are not different based on t-test at significant level 0.05.

| Methods | ChestX | | ISIC | |
|---|---|---|---|---|
| | k=1 | k=5 | k=1 | k=5 |
| MAML* | - | $23.48 \pm 0.96$ | - | $40.13 \pm 0.58$ |
| ProtoNet* | - | $24.05 \pm 1.01$ | - | $39.57 \pm 0.57$ |
| ProtoNet + FWT* | - | $23.77 \pm 0.42$ | - | $38.87 \pm 0.52$ |
| MetaOpt* | - | $22.53 \pm 0.91$ | - | $36.28 \pm 0.50$ |
| Transfer* | - | $25.35 \pm 0.96$ | - | $43.56 \pm 0.60$ |
| Transfer | $22.71 \pm 0.40$ | $\mathbf{26.71 \pm 0.46}$ | $30.71 \pm 0.59$ | $43.08 \pm 0.57$ |
| SimCLR | $22.10 \pm 0.41$ | $25.02 \pm 0.42$ | $26.25 \pm 0.53$ | $36.09 \pm 0.57$ |
| Transfer + SimCLR | $22.70 \pm 0.40$ | $\mathbf{26.95 \pm 0.45}$ | $\mathbf{32.63 \pm 0.63}$ | $45.96 \pm 0.61$ |
| STARTUP-T (no SS) | $\mathbf{22.79 \pm 0.41}$ | $26.03 \pm 0.43$ | $\mathbf{32.37 \pm 0.61}$ | $45.20 \pm 0.61$ |
| STARTUP (no SS) | $\mathbf{22.87 \pm 0.41}$ | $\mathbf{26.68 \pm 0.45}$ | $\mathbf{32.24 \pm 0.62}$ | $46.48 \pm 0.61$ |
| STARTUP-T | $22.75 \pm 0.40$ | $26.47 \pm 0.43$ | $32.16 \pm 0.60$ | $45.75 \pm 0.60$ |
| STARTUP | $\mathbf{23.09 \pm 0.43}$ | $\mathbf{26.94 \pm 0.44}$ | $\mathbf{32.66 \pm 0.60}$ | $\mathbf{47.22 \pm 0.61}$ |
| STARTUP (w/ Rotation) | $\mathbf{22.83 \pm 0.42}$ | $26.38 \pm 0.43$ | $31.54 \pm 0.61$ | $45.68 \pm 0.60$ |

| Methods | EuroSAT | | CropDisease | |
|---|---|---|---|---|
| | k=1 | k=5 | k=1 | k=5 |
| MAML* | - | $71.70 \pm 0.72$ | - | $78.05 \pm 0.68$ |
| ProtoNet* | - | $73.29 \pm 0.71$ | - | $79.72 \pm 0.67$ |
| ProtoNet + FWT* | - | $67.34 \pm 0.76$ | - | $72.72 \pm 0.70$ |
| MetaOpt* | - | $64.44 \pm 0.73$ | - | $68.41 \pm 0.73$ |
| Transfer* | - | $75.69 \pm 0.66$ | - | $87.48 \pm 0.58$ |
| Transfer | $60.73 \pm 0.86$ | $80.30 \pm 0.64$ | $69.97 \pm 0.85$ | $90.16 \pm 0.49$ |
| SimCLR | $43.52 \pm 0.88$ | $59.05 \pm 0.70$ | $\mathbf{78.23 \pm 0.83}$ | $92.57 \pm 0.48$ |
| Transfer + SimCLR | $57.18 \pm 0.87$ | $77.61 \pm 0.66$ | $76.90 \pm 0.78$ | $92.64 \pm 0.44$ |
| STARTUP-T (no SS) | $63.00 \pm 0.84$ | $81.25 \pm 0.62$ | $71.11 \pm 0.83$ | $90.79 \pm 0.49$ |
| STARTUP (no SS) | $62.90 \pm 0.83$ | $81.81 \pm 0.61$ | $73.30 \pm 0.82$ | $91.69 \pm 0.47$ |
| STARTUP-T | $63.49 \pm 0.85$ | $81.54 \pm 0.63$ | $72.85 \pm 0.83$ | $91.49 \pm 0.48$ |
| STARTUP | $\mathbf{63.88 \pm 0.84}$ | $\mathbf{82.29 \pm 0.60}$ | $75.93 \pm 0.80$ | $\mathbf{93.02 \pm 0.45}$ |
| STARTUP (w/ Rotation) | $62.18 \pm 0.86$ | $81.37 \pm 0.65$ | $70.53 \pm 0.84$ | $90.59 \pm 0.48$ |

Table 6: 5-way k-shot classification accuracy on miniImageNet→BSCD-FSL for higher shots. Mean and 95% confidence interval are reported. $^*$ are methods reported in (Guo et al., 2020). Despite using their code, difference in batch size and test set (80% of the original test set) have resulted in discrepancies between our Transfer and their Transfer$^*$. (no SS) indicates removal of SimCLR. Bolded entries are top performing methods that are not different based on t-test at significant level 0.05.

| Methods | ChestX | | ISIC | |
| --- | --- | --- | --- | --- |
| | k=20 | k=50 | k=20 | k=50 |
| MAML$^*$ | $27.53 \pm 0.43$ | - | $52.36 \pm 0.57$ | - |
| ProtoNet$^*$ | $28.21 \pm 1.15$ | $29.32 \pm 1.12$ | $49.50 \pm 0.55$ | $51.99 \pm 0.52$ |
| ProtoNet + FWT$^*$ | $26.87 \pm 0.43$ | $30.12 \pm 0.46$ | $43.78 \pm 0.47$ | $49.84 \pm 0.51$ |
| MetaOpt$^*$ | $25.53 \pm 1.02$ | $29.35 \pm 0.99$ | $49.42 \pm 0.60$ | $54.80 \pm 0.54$ |
| Transfer$^*$ | $30.83 \pm 1.05$ | $36.04 \pm 0.46$ | $52.78 \pm 0.58$ | $57.34 \pm 0.56$ |
| Transfer | $31.99 \pm 0.46$ | $35.74 \pm 0.47$ | $54.28 \pm 0.59$ | $60.26 \pm 0.56$ |
| SimCLR | $29.62 \pm 0.44$ | $32.69 \pm 0.42$ | $47.17 \pm 0.58$ | $52.55 \pm 0.56$ |
| Transfer + SimCLR | $32.73 \pm 0.46$ | $\mathbf{36.64 \pm 0.47}$ | $57.33 \pm 0.59$ | $62.84 \pm 0.60$ |
| STARTUP-T (no SS) | $31.77 \pm 0.44$ | $35.57 \pm 0.47$ | $55.80 \pm 0.59$ | $61.15 \pm 0.56$ |
| STARTUP (no SS) | $\mathbf{33.02 \pm 0.47}$ | $\mathbf{36.72 \pm 0.47}$ | $57.41 \pm 0.57$ | $62.71 \pm 0.56$ |
| STARTUP-T | $32.79 \pm 0.46$ | $\mathbf{36.66 \pm 0.47}$ | $56.43 \pm 0.60$ | $61.76 \pm 0.58$ |
| STARTUP | $\mathbf{33.19 \pm 0.46}$ | $\mathbf{36.91 \pm 0.50}$ | $\mathbf{58.63 \pm 0.58}$ | $\mathbf{64.16 \pm 0.58}$ |
| STARTUP (w/ Rotation) | $31.94 \pm 0.47$ | $36.33 \pm 0.46$ | $57.97 \pm 0.60$ | $63.44 \pm 0.57$ |

| Methods | EuroSAT | | CropDisease | |
| --- | --- | --- | --- | --- |
| | k=20 | k=50 | k=20 | k=50 |
| MAML$^*$ | $81.95 \pm 0.55$ | - | $89.75 \pm 0.42$ | - |
| ProtoNet$^*$ | $82.27 \pm 0.57$ | $80.48 \pm 0.57$ | $88.15 \pm 0.51$ | $90.81 \pm 0.43$ |
| ProtoNet + FWT$^*$ | $75.74 \pm 0.70$ | $78.64 \pm 0.57$ | $85.82 \pm 0.51$ | $87.17 \pm 0.50$ |
| MetaOpt$^*$ | $79.19 \pm 0.62$ | $83.62 \pm 0.58$ | $82.89 \pm 0.54$ | $91.76 \pm 0.38$ |
| Transfer$^*$ | $84.13 \pm 0.52$ | $86.62 \pm 0.47$ | $94.45 \pm 0.36$ | $96.62 \pm 0.25$ |
| Transfer | $88.31 \pm 0.45$ | $91.09 \pm 0.37$ | $96.10 \pm 0.28$ | $97.69 \pm 0.20$ |
| SimCLR | $72.25 \pm 0.58$ | $78.64 \pm 0.50$ | $97.26 \pm 0.23$ | $\mathbf{98.44 \pm 0.17}$ |
| Transfer + SimCLR | $87.48 \pm 0.45$ | $91.43 \pm 0.35$ | $\mathbf{97.41 \pm 0.22}$ | $\mathbf{98.44 \pm 0.17}$ |
| STARTUP-T (no SS) | $88.44 \pm 0.46$ | $91.13 \pm 0.38$ | $96.31 \pm 0.28$ | $97.80 \pm 0.20$ |
| STARTUP (no SS) | $\mathbf{89.29 \pm 0.43}$ | $\mathbf{91.94 \pm 0.35}$ | $96.91 \pm 0.25$ | $98.20 \pm 0.17$ |
| STARTUP-T | $88.39 \pm 0.46$ | $91.27 \pm 0.38$ | $96.69 \pm 0.26$ | $98.05 \pm 0.19$ |
| STARTUP | $\mathbf{89.26 \pm 0.43}$ | $\mathbf{91.99 \pm 0.36}$ | $\mathbf{97.51 \pm 0.21}$ | $\mathbf{98.45 \pm 0.17}$ |
| STARTUP (w/ Rotation) | $88.78 \pm 0.46$ | $91.63 \pm 0.37$ | $96.49 \pm 0.27$ | $98.01 \pm 0.19$ |

Table 7: 5-way k-shot classification accuracy on ImageNet(ILSVRC 2012)→BSCD-FSL. Bolded entries are top performing methods that are not different based on t-test at significant level 0.05.

| Methods | ChestX | | ISIC | |
|---|---|---|---|---|
| | k=1 | k=5 | k=1 | k=5 |
| Transfer | $21.97 \pm 0.39$ | $25.85 \pm 0.41$ | $30.27 \pm 0.51$ | $43.88 \pm 0.56$ |
| STARTUP (no SS) | $\mathbf{22.90 \pm 0.40}$ | $26.74 \pm 0.46$ | $30.18 \pm 0.56$ | $44.19 \pm 0.57$ |
| STARTUP | $\mathbf{23.03 \pm 0.42}$ | $\mathbf{27.24 \pm 0.46}$ | $\mathbf{31.69 \pm 0.59}$ | $\mathbf{46.02 \pm 0.59}$ |
| Methods | EuroSAT | | CropDisease | |
| | k=1 | k=5 | k=1 | k=5 |
| Transfer | $66.08 \pm 0.81$ | $85.58 \pm 0.48$ | $74.17 \pm 0.82$ | $92.46 \pm 0.42$ |
| STARTUP (no SS) | $70.08 \pm 0.80$ | $87.12 \pm 0.45$ | $80.13 \pm 0.77$ | $94.51 \pm 0.38$ |
| STARTUP | $\mathbf{73.83 \pm 0.77}$ | $\mathbf{89.70 \pm 0.41}$ | $\mathbf{85.10 \pm 0.74}$ | $\mathbf{96.06 \pm 0.33}$ |

Table 8: 5-way k-shot classification accuracy on ImageNet(ILSVRC 2012)→BSCD-FSL for higher shots. Bolded entries are top performing methods that are not different based on t-test at significant level 0.05.

| Methods | ChestX | | ISIC | |
|---|---|---|---|---|
| | k=20 | k=50 | k=20 | k=50 |
| Transfer | $30.28 \pm 0.45$ | $32.55 \pm 0.46$ | $55.14 \pm 0.60$ | $60.99 \pm 0.60$ |
| STARTUP (no SS) | $\mathbf{31.98 \pm 0.47}$ | $34.22 \pm 0.47$ | $55.54 \pm 0.57$ | $61.54 \pm 0.55$ |
| STARTUP | $\mathbf{32.40 \pm 0.45}$ | $\mathbf{34.95 \pm 0.48}$ | $\mathbf{57.06 \pm 0.58}$ | $\mathbf{62.94 \pm 0.56}$ |
| Methods | EuroSAT | | CropDisease | |
| | k=20 | k=50 | k=20 | k=50 |
| Transfer | $91.78 \pm 0.33$ | $93.76 \pm 0.29$ | $96.96 \pm 0.25$ | $98.10 \pm 0.19$ |
| STARTUP (no SS) | $92.60 \pm 0.31$ | $94.53 \pm 0.26$ | $97.94 \pm 0.20$ | $98.62 \pm 0.16$ |
| STARTUP | $\mathbf{94.27 \pm 0.26}$ | $\mathbf{95.61 \pm 0.23}$ | $\mathbf{98.55 \pm 0.17}$ | $\mathbf{99.07 \pm 0.13}$ |

Table 9: 5-way k-shot classification accuracy on miniImageNet→BSCD-FSL for different initialization strategies. Mean and 95% confidence interval are reported. Bolded entries are top performing methods that are not different based on t-test at significant level 0.05.

| Methods | ChestX | | ISIC | |
|---|---|---|---|---|
| | k=1 | k=5 | k=1 | k=5 |
| STARTUP-Rand (no SS) | $22.38 \pm 0.41$ | $24.96 \pm 0.41$ | $29.76 \pm 0.60$ | $40.45 \pm 0.59$ |
| STARTUP-T (no SS) | $\mathbf{22.79 \pm 0.41}$ | $26.03 \pm 0.43$ | $\mathbf{32.37 \pm 0.61}$ | $45.20 \pm 0.61$ |
| STARTUP (no SS) | $\mathbf{22.87 \pm 0.41}$ | $26.68 \pm 0.45$ | $\mathbf{32.24 \pm 0.62}$ | $\mathbf{46.48 \pm 0.61}$ |
| Methods | EuroSAT | | CropDisease | |
| | k=1 | k=5 | k=1 | k=5 |
| STARTUP-Rand (no SS) | $\mathbf{63.44 \pm 0.89}$ | $81.05 \pm 0.60$ | $\mathbf{74.44 \pm 0.83}$ | $\mathbf{92.04 \pm 0.47}$ |
| STARTUP-T (no SS) | $\mathbf{63.00 \pm 0.84}$ | $81.25 \pm 0.62$ | $71.11 \pm 0.83$ | $90.79 \pm 0.49$ |
| STARTUP (no SS) | $62.90 \pm 0.83$ | $\mathbf{81.81 \pm 0.61}$ | $73.30 \pm 0.82$ | $91.69 \pm 0.47$ |

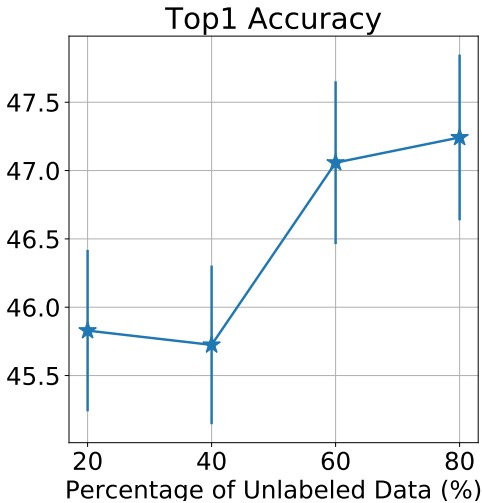

Figure 3: 5-way 5-shot Classification Accuracy of STARTUP for miniImageNet → ISIC with various amount of unlabeled data. Mean and 95% confidence interval over 600 tasks are plotted.

Table 10: 5-way k-shot classification accuracy on miniImageNet→BSCD-FSL. We compare STARTUP to fine-tuning. Bolded entries are top performing methods that are not different based on t-test at significant level 0.05.

| Methods | ChestX | | ISIC | |
|---|---|---|---|---|
| | k=1 | k=5 | k=1 | k=5 |
| STARTUP | $\textbf{23.09} \pm \textbf{0.43}$ | $\textbf{26.94} \pm \textbf{0.44}$ | $\textbf{32.66} \pm \textbf{0.60}$ | $\textbf{47.22} \pm \textbf{0.61}$ |
| Fine-tuning | $22.76 \pm 0.41$ | $\textbf{27.05} \pm \textbf{0.45}$ | $\textbf{32.45} \pm \textbf{0.61}$ | $45.73 \pm 0.61$ |

| Methods | EuroSAT | | CropDisease | |
|---|---|---|---|---|
| | k=1 | k=5 | k=1 | k=5 |
| STARTUP | $\textbf{63.88} \pm \textbf{0.84}$ | $\textbf{82.29} \pm \textbf{0.60}$ | $\textbf{75.93} \pm \textbf{0.80}$ | $\textbf{93.02} \pm \textbf{0.45}$ |
| Fine-tuning | $62.86 \pm 0.85$ | $\textbf{82.36} \pm \textbf{0.61}$ | $\textbf{76.13} \pm \textbf{0.78}$ | $\textbf{93.01} \pm \textbf{0.44}$ |

Table 11: Few-shot classification accuracy on ImageNet(ILSVRC 2012)→BSCD-FSL with STARTUP on different datasets. STARTUP-X represents the STARTUP student trained on ImageNet and dataset X. The top table presents the results for 5-way 1-shot and the bottom table presents the results for 5-way 5-shot. Bolded entries are top performing methods that are not different based on t-test at significant level 0.05.

| Methods | ChestX | ISIC | EuroSAT | CropDisease |
|---|---|---|---|---|
| STARTUP-ChestX | $\textbf{23.03} \pm \textbf{0.42}$ | $31.02 \pm 0.55$ | $65.20 \pm 0.87$ | $70.36 \pm 0.86$ |
| STARTUP-ISIC | $21.98 \pm 0.39$ | $\textbf{31.69} \pm \textbf{0.59}$ | $61.31 \pm 0.78$ | $69.27 \pm 0.87$ |
| STARTUP-EuroSAT | $22.23 \pm 0.38$ | $30.38 \pm 0.59$ | $\textbf{73.83} \pm \textbf{0.77}$ | $70.40 \pm 0.86$ |
| STARTUP-CropDisease | $22.51 \pm 0.40$ | $30.59 \pm 0.55$ | $62.56 \pm 0.83$ | $\textbf{85.10} \pm \textbf{0.74}$ |
| Methods | ChestX | ISIC | EuroSAT | CropDisease |
| STARTUP-ChestX | $\textbf{27.24} \pm \textbf{0.46}$ | $44.14 \pm 0.59$ | $83.76 \pm 0.56$ | $89.95 \pm 0.49$ |
| STARTUP-ISIC | $25.05 \pm 0.42$ | $\textbf{46.02} \pm \textbf{0.59}$ | $81.57 \pm 0.55$ | $89.24 \pm 0.50$ |
| STARTUP-EuroSAT | $25.15 \pm 0.43$ | $43.90 \pm 0.58$ | $\textbf{89.70} \pm \textbf{0.41}$ | $88.78 \pm 0.54$ |
| STARTUP-CropDisease | $25.21 \pm 0.44$ | $44.34 \pm 0.60$ | $83.13 \pm 0.54$ | $\textbf{96.06} \pm \textbf{0.33}$ |

