# OpenReview forum: "Self-training For Few-shot Transfer Across Extreme Task Differences"
_ICLR.cc/2021/Conference — ICLR 2021 Oral_

### Official Review · AnonReviewer2 · 2020-10-27
**SimCLR based co-training for extreme domain transfer**

**Rating:** 7
**Confidence:** 4

**Review:**

This nicely written paper explores the situation in which one tries to do domain transfer from a large dataset with many labels to a partially labeled dataset with relatively few unlabeled and even less labeled examples, if the domain gap is very large.

The paper presents a pre-training methodology in which the dataset is first co-trained on the large, labeled dataset and the unlabeled examples. They use the SimCLR objective for pretraining on the unlabeled examples.

This situation explored here is extremely common in practice, especially in medical imaging: even unlabeled training data is relatively rare, however, it is even rarer to have high quality training data.

This approach show premise in tackling the domain transfer problem from image-net images to chest X-rays. The paper reports significant progress on chest-ray images and therefore providing important practical contributions in this domain.

While the novelty of the approach is somewhat limited: it just combines an existing pretraining approach with co-training on another dataset, its simplicity makes up for it. Also, I am not aware of any work that tackles this important issue so elegantly and directly.

Also the paper presents two hypothesis on why this approach works. One just posits that the additionally unsupervised data helps independently of the target domain. The second posits that the pretraining helps grouping in the target domain. The paper presents convincing ablation analysis to back the latter claim: training on the target domain is helpful for transferring to that specific domain.

Strength:
- High quality writing. Easy to read paper
- Good practical results
- Tackles a practically highly relevant problem
- Convincing ablation analyses
- Simplicity of the approach

Weaknesses:
- Somewhat limited novelty
- No theoretical analysis
- Evaluated on a single scenario

Despite the fact that the approach is somewhat straightforward and of limited novelty and while it was evaluated on one particular use-case, the approach seems to be highly relevant in practice, it is relatively simple and practical and exhibits good performance in an extremely important scenario.

---

### Official Review · AnonReviewer1 · 2020-10-27
**Good start; but not good enough yet.**

**Rating:** 6
**Confidence:** 4

**Review:**

**Summary of the paper**

This paper studies transfer learning in an episode learning setting, where at evaluation time a few-shot few-example task is generated. In contrast to the standard setting, two modifications are made:
1) large domain differences (base dataset is (mini)ImageNet, target datasets are plant crops, xrays, satellite images) from the BSCD-FSL benchmark (Guo et al., 2020);
2) the learner is given unlabeled target images for domain adaptation.
The technical novelty of the paper is to use the unlabeled dataset by (a) obtaining soft-labels from miniImageNet classes, and (b) using SimCLR (Chen et al, 2020) as self-supervised loss.

**Strengths**

This setting is interesting especially due to the large domain differences. Adding self-supervision under large domain shift seems a logical choice.

**Weaknesses**

- There is no comparison to other self-supervised methods / unsupervised domain adaptation methods for few-shot learning.  In this paper only a single self-supervised method (SimCLR) is evaluated, while there are other methods also focussing on self-supervision for few-shot learning (Gidaris et al., 2019, Su et al., 2020). While these approaches are not evaluated on the large domain shift, does not tell anything about the relevance for this setting.
- There is no comparison on a standard benchmark, while a benchmark dataset is used, the benchmark evaluation is omitted, only per dataset / task scores are provided.
- There is no comparison on other domain transfer tasks, for example on more standard MAML types of domain transfers. To allow to compare to other self-supervised methods for few-shot learning.
- The title might be misleading: Self-Training might no do so much, according to table 1. The performance difference between Transfer and SimCLR baselines are either about equal (ChestX, CropDisease) or Transfer is much better (ISIC +7, EuroSAT +21). Startup shows a small (in the range +.2 -  +4) performance increase over Transfer, which might suggest that SimCLR acts only as a small regularizer. This is confirmed by the ablation of different startup strategies. Unfortunately, the plain combination of Transfer + SimCLR is not explored.

**Summary**

This paper presents an interesting line of research on few-shot learning on large domain shifts. The technical novelty is minor: it propose to use SimCLR as auxiliary loss in a student-teacher setup. Without comparing to other work or related methods. Experimentally this setup is evaluated on the BSCD-FSL benchmark dataset (yet, without using the benchmark evaluation). Therefore I give an 'OK but not good enough' rating. It seems a good start, but more comparisons, and evaluations are needed for publication.

**Post Rebuttal**
My main concerns for the paper are not withdrawn, yet not shared with the other reviewers. I think that is fine, it shows that perspectives differ, and that it is in part why multiple reviewers should read a paper. I also do see that the paper has been improved based on the provided feedback. Given that the paper does not contain major flaws, I upgrade my vote to 6.

---

### Official Review · AnonReviewer4 · 2020-10-29

**Rating:** 8
**Confidence:** 5

**Review:**

#### Summary

The submission tackles the problem of cross-domain few-shot learning in a setting where unlabeled data is available for the test domains.

It introduces an approach called "Self Training to Adapt Representations to Unseen Problems" (STARTUP) which first pre-trains a teacher model on the (labeled) base dataset, and then distills the teacher model into a student model (initialized with the teacher parameters and a random output layer) on the base set of classes as well as the (unlabeled) target domain dataset. For the target domain examples, the loss is a combination of the KL-divergence between logits output by the student and teacher models and an additional unsupervised/self-supervised loss function (SimCLR in this instance). For test episodes, STARTUP is free to choose any applicable inference approach, and the paper opts to fit a linear classifier on top of the frozen feature extractor.

The proposed approach is evaluated on BSCD-FSL using a random 20% subset of each target domain to form unlabeled datasets and the remaining 80% to form evaluation episodes. STARTUP is compared with recent cross-domain few-shot classification approaches (which do not use unlabeled target domain data) and a purely self-supervised baseline which ignores the base dataset and learns a representation from the target domain using SimCLR.

The submission also investigates two hypotheses to explain why STARTUP improves performance, and concludes that its purpose is *not* to simply introduce noise during training and instead appears to emphasize the natural groupings induced by the base classifier on the target domain data.

#### Strengths and weaknesses

* **+** Clarity: the paper is well-written and easy to follow.
* **+** Cross-domain few-shot classification is very relevant to the few-shot learning research community, and the availability of unlabeled data for the test domains is a plausible assumption.
* **+** The proposed approach is sound and straightforward.
* **+** The paper makes an effort at explaining why STARTUP provides a performance improvement.
* **-** The paper should be more rigorous when reporting which approach performs best in a given setting.

#### Recommendation

I recommend acceptance. Overall the paper is clear, the problem being tackled is relevant to the research community, the proposed idea is sound, and evaluation is rigorous. My main concern has to do with the way in which best-performing approaches are reported, but that’s easily fixable in a subsequent version of the paper.

#### Detailed justification

I appreciate the quality of the writing. The proposed idea is straightforward and makes intuitive sense. The baselines chosen for comparison are reasonable.

The main concern I have is with the way in which results are reported in Table 1. I believe the authors bolded the best-performing entry in each setting without taking the 95% confidence intervals into account. As an example, can we say that STARTUP performs significantly better than Transfer in the ChestX 5-way 1-shot/5-shot settings when their 95% confidence intervals overlap as much as they do? In my opinion the more rigorous way to determine that would be to run a 95% confidence statistical test on the difference between the means and bold all entries for which the test is inconclusive in rejecting the hypothesis that the difference in mean to the best-performing entry is zero, like is done in the Meta-Dataset paper. This doesn’t change the main conclusions drawn by the paper, but should nevertheless be addressed.

#### Questions

1. When training the student model, would there be a benefit to compute the loss as the KL-divergence between the label distributions output by the teacher and student models instead of the cross-entropy with the true label? Have the authors investigated this?
2. The proposed approach has the downside that it requires training and storing a separate student model for each target domain. Have the authors thought of more compact approaches, like domain-conditional self-training?

#### Additional feedback

1. The abstract mentions evaluating on a "challenging benchmark with multiple domains" but does not name it (BSCD-FSL). This is information that would be helpful to the reader.
1. The insistence on "several hours of compute" in the introduction as being a drawback of current recognition systems could be toned down. To me, training a model for several hours doesn’t sound unreasonable.
1. The Visual Task Adaptation Benchmark (VTAB) and Big Transfer (BiT) papers would be relevant to mention in the related work section. In particular, the VTAB paper investigates various representation learning strategies (including self-supervision) when transferring between very different base and novel domains.
1. The strategy to use logits on the label set of the base dataset as targets (and therefore leveraging the natural groupings induced by the base dataset classifier) reminds me in a way of Ngiam et al. (2018)’s work on "Domain Adaptive Transfer Learning with Specialist Models", which uses this to inform the weighting of classes used to pre-train a target domain-aware classifier on the base dataset. This paper’s proposed approach differs in many ways, but the similarities would be interesting to expand upon.

---

### Official Review · AnonReviewer3 · 2020-11-01
**Exciting Setup, Interesting Experiments -- Concerns with the Conclusion**

**Rating:** 8
**Confidence:** 5

**Review:**

Problem: The paper introduces the problem of few-shot transfer when there is an extreme difference between the base task and the target task.  The usual few-shot learning setup considers a representation that is trained on a large amount of labeled data. This base representation is then fine-tuned for the target task (that has a few examples, say 1 or 5 labeled examples per class). This strategy works well when the data distribution of the base and target task is similar. However, few-shot learners fail when the data distribution for the two domains are different (e.g., imagenet and crop-diseases) as shown by Guo et al., 2020.

Solution: In this work, authors intuit that there may be a large amount of unlabeled data for the target domain that can be used to learn a good representation of the target task. This is done by using a teacher model trained on a large dataset to generate soft-labels on the unlabeled dataset. A new model is trained using: (1) base dataset; (2) soft-labels on unlabeled data; and (3) contrastive learning on unlabeled data. The new model is initialized using the base representation of the teacher model and the classifier (last layer) is trained from scratch. At test time, *this new representation* is used for learning a classifier on the few labeled examples. This approach has been demonstrated to improve the performance by an average 2.9 points on four tasks: (1) X-Ray; (2) EuroSat; (3) Crop Diseases; and (4) ISIC.

Pros:

+ simple approach in yielding better performance!

+ use of unlabeled data from the target domain to learn a better representation.

+ while authors position themselves as "self-training" -- I think it is not self-training and is basically about "conditioning the existing representation on the target task". To the best of my knowledge, the authors have a very unique and interesting perspective that I have not seen previously.

Cons:

Following are my concerns with the submission:

1. Chest X-Ray and ISIC dataset vs. Crop-Disease and Satellite Images - The proposed approach works for Crop Disease and Satellite Images which are partly in the domain of base task. It does not show improvement for Chest X-Ray and ISIC dataset that is extremely different from the base task. Is it not against the premise of this paper (specifically the first paragraph of Sec.1)?

2. Section - 4 (Training a new student model): The optimization consists of three parts: (a) supervised loss using base task; (b) pseudo-labels on target task; and (c) contrastive learning on unlabeled images. The evaluation does not show a meaningful difference in the use of these losses except for Crop Diseases dataset.

Note: The authors mention the similarity of (a) and (b) in optimization used in Xie at al., 2020. The statement is nuanced. Strictly speaking, Xie et al., 2020 did not use (a) and (b). They rather trained using the hard pseudo-labels generated on the unlabeled images (using the teacher model).

3. The student model is initialized using the base representation and classifier from scratch. Different initialization strategies are explored. However, conclusions in Section 4.2 are not in sync with Table 7 in Appendix.

4. Section-5.2.1: Hypothesis-1 -- The conclusion of Xie et al., 2020 are valid for their setup (which is strictly for ImageNet classification).

5. Section-5.3: This experiment is inconclusive because of the limited unlabeled data used here.

Now zooming-out and some high-level concerns:

1. First line of the Abstract: "All few-shot learning techniques *must* be pre-trained on large, labeled dataset." --> Why?

I can buy that currently all few-shot learning techniques *are* pre-trained on large, labeled dataset, but I don't see why it is a "must"?

2. First paragraph of Section.1: What has few-shot learning to do with computational time? If it is true, then the proposed formulation does not come any close to solving it because it requires training a new model on a large labeled and unlabeled dataset for a new target task.

POST-REBUTTAL: I have updated the score from 5 to 7 following the clarifications from the authors.

---

### Decision · Program_Chairs · 2021-01-07
**Final Decision**

**Decision:**

Accept (Oral)

**Comment:**

The paper introduces an approach to self-train a source domain classifier on unlabeled data from the target domain, considering the few-shot learning setting when there is significant discrepancy between the source and target domains. While the reviewers pointed out a few weaknesses, such as somewhat limited methodological novelty  and lack of comparisons with other methods, they all recommend acceptance as final decision. The paper is beautifully written. The proposed method is very simple, but yields excellent results in a very practical problem, which should be of wide interest to the ICLR community. The experimental evaluation is rigorous and the ablation studies are convincing. The AC agrees with the decision made by the reviewers and recommends acceptance.